**PLOS** | COMPUTATIONAL BIOLOGY

# Doubting what you already know: Uncertainty regarding state transitions is associated with obsessive compulsive symptoms

Isaac Fradkin[1]*, Casimir Ludwig[2], Eran Eldar[1,3], Jonathan D. Huppert[1]

1 The Hebrew University of Jerusalem, Mount Scopus, Jerusalem, Israel, 2 School of Psychological Science, University of Bristol, Bristol, United Kingdom, 3 Max Planck-UCL Center for Computational Psychiatry and Ageing Research, London United Kingdom

* itzik.fradkin@mail.huji.ac.il

**Data Availability Statement:** The data and computational models can be found in a public repository for review: http://doi.org/10.17605/OSF.IO/D6B3M.

## Abstract

Obsessive compulsive (OC) symptoms involve excessive information gathering (e.g., checking, reassurance-seeking), and uncertainty about possible, often catastrophic, future events. Here we propose that these phenomena are the result of *excessive uncertainty regarding state transitions (transition uncertainty)*: a computational impairment in Bayesian inference leading to a reduced ability to use the past to predict the present and future, and to oversensitivity to feedback (i.e. prediction errors). Using a computational model of Bayesian learning under uncertainty in a reversal learning task, we investigate the relationship between OC symptoms and transition uncertainty. Individuals high and low in OC symptoms performed a task in which they had to detect shifts (i.e. transitions) in cue-outcome contingencies. Modeling subjects' choices was used to estimate each individual participant's transition uncertainty and associated responses to feedback. We examined both an optimal observer model and an approximate Bayesian model in which participants were assumed to attend (and learn about) only one of several cues on each trial. Results suggested the participants were more likely to distribute attention across cues, in accordance with the optimal observer model. As hypothesized, participants with higher OC symptoms exhibited increased transition uncertainty, as well as a pattern of behavior potentially indicative of a difficulty in relying on learned contingencies, with no evidence for perseverative behavior. Increased transition uncertainty compromised these individuals' ability to predict ensuing feedback, rendering them more surprised by expected outcomes. However, no evidence for excessive belief updating was found. These results highlight a potential computational basis for OC symptoms and obsessive compulsive disorder (OCD). The fact the OC symptoms predicted a decreased reliance on the past rather than perseveration challenges preconceptions of OCD as a disorder of inflexibility. Our results have implications for the understanding of the neurocognitive processes leading to excessive uncertainty and distrust of past experiences in OCD.

**Funding:** Preparation of this manuscript was supported by the Israel Science Foundation (https://www.isf.org.il/); grant #1698/15 to JDH. The funder did not play any role in study design, data collection, analysis, decision to publish or preparation of the manuscript

**Competing interests:** The authors have declared that no competing interests exist.

## Author summary

Obsessive compulsive (OC) symptoms involve excessive information gathering (e.g., checking, reassurance seeking), and excessive uncertainty about possible future events. Normally, people can use prior experience to predict present and future events. Here we suggest that OC symptoms can be traced back to an impairment in this prediction mechanism. In Bayesian models of learning and decision making the relative weight given to prior experience depends on the estimation of uncertainty. Particularly, when one believes that past states cannot predict the future with certainty, the optimal behavior is to assign a higher weight to current feedback at the expense of prior experience. We examined this mechanism, using a task that required participants to learn cue-outcome contingencies from feedback, while considering the possibility that occasional changes in the contingencies render past experience irrelevant. A computational analysis of participants' behavior showed that participants with higher OC symptoms indeed assigned lower weight to prior experience, leading to over-exploratory behavior. These results have implications for the understanding of the neurocognitive processes leading to excessive uncertainty and distrust of past experiences in obsessive compulsive disorder.

## Introduction

Imagine that you place your wallet into your bag. Normally this behavior, often automatic, would allow you feel confident that your wallet is there. However, if you happen to know that your bag has a hole in it, you will be uncertain that your wallet will stay in your bag because the wallet's past state (i.e., in bag) cannot reliably predict its present state. Therefore, you will be more likely to worry about your wallet falling out, trying to prevent this from happening or constantly checking that your wallet is still there.

Obsessive compulsive (OC) symptoms often involve such preemptive actions and checking behavior. Patients and subclinical populations with elevated OC symptoms exhibit similar behavior even in experimental contexts that do not activate OC-related fears, suggesting that a basic cognitive function might be impaired. Indeed, obsessive compulsive disorder (OCD) and OC symptoms have been associated with longer search times and more fixations in visual search tasks [1,2], and more repetitive checking behavior in change detection tasks [3,4], potentially implying decreased utilization of previously accumulated information. More specific evidence comes from a recent study using a complex probabilistic learning task which showed that OCD patients failed to make full use of previously accumulated knowledge about the environment, such that their behavior excessively reflected the most recent observations [5]. Indeed, patients' difficulty in trusting their own memory [6], and tendency to repeatedly doubt and re-examine what they should already know (e.g., that the stove is already off; that one's hands are already clean) might be related to a cognitive impairment in relying upon accumulated knowledge.

Conversely, numerous studies have pursued the idea that OCD is characterized by cognitive inflexibility and perseveration: a difficulty in forsaking learned contingencies or responses [7–9]. This is often examined in reversal learning tasks, wherein participants are required to adapt to changes in task contingencies. Notably, this idea stands in stark contrast to the idea of decreased reliance on previous knowledge in OCD, articulated above. However, a recent meta-analysis of flexibility in OCD showed that patients' behavior in such tasks does not evidence a specific pattern of perseveration, but instead is best characterized as non-specific underperformance [7]. Most behavioral indices used in such tasks are likely governed by a

complex interaction of different cognitive processes, which might lead to the appearance of global underperformance. In the current study we use a computational modeling approach designed to uncover the specific cognitive processes (rather than global behavioral measures) that correlate with OC symptoms in a reversal learning task.

Reversal learning tasks require participants to use feedback to learn which of several cues is currently advantageous. Before a shift in contingencies occurs, participants can rely on previously accumulated knowledge, and ignore current feedback. However, because participants do not know a-priori when such a shift will occur, they must consider both current feedback and previous knowledge. Furthermore, if outcomes are determined probabilistically (i.e., feedback is not fully reliable), as in probabilistic reversal learning tasks, participants are required to decide whether unexpected feedback is misleading or indicates a real contingency shift.

In line with the influential idea that the brain implements some sort of Bayesian inference [10,11], this intuitive process can be formalized in a Bayesian state-space model that aims to infer the current state of the environment [12]. In the context of reversal learning this corresponds to inferring which cue is currently advantageous. Bayesian inference provides a principled way of integrating prior knowledge and current evidence by weighting each by its relative uncertainty [10,13–15]. In particular, learning is governed by the balance between two types of uncertainty: uncertainty regarding state transitions (i.e., transition uncertainty) and observation uncertainty. The former (inversely) reflects the belief that past evidence (i.e. state at t-1) is predictive of the current state (at time t; e.g., the expectation that my wallet is in my bag if it was there before; the expectation that the previously advantageous cue is still advantageous). The latter reflects the belief that current feedback faithfully reflects the current state (e.g., can sensory feedback indicate the location of my wallet? How reliable is the current feedback with regards to which cue is advantageous?), and is especially relevant in probabilistic reversal learning. Similar uncertainty-related processes are involved in models postulating that the brain is only approximating Bayesian inference [13].

The use of this computational formalization allows us to test hypotheses regarding the cognitive processes underlying readily apparent behavioral manifestation such as perseveration or poor performance. Thus, perseveration (i.e., disregard of feedback indicating a contingency shift) can result from at least two processes: overreliance on previous knowledge (i.e., underestimation of transition uncertainty), or under-reliance on current feedback (i.e., overestimation of observation uncertainty). However, as suggested above, it is also possible that OC symptoms are actually rooted in excessive transition uncertainty–leading to disregard of previous knowledge, and inducing repetitive seeking of new information (manifesting as checking, reassurance seeking, etc.) that is then given excessive (but short-lived) weight in shaping one's beliefs. Indeed, a recent computational account of OCD suggests that obsessive compulsive pathology can be traced back to excessive transition uncertainty [12]. Modeling can arbitrate between these possibilities, while also examining the possibility that poor performance reflects non-specific random responding [7] due to a more trivial cause such as inattention or a lack of motivation. We examine these questions using an adapted version of the reversal learning task proposed by Yu and Dayan [13], which allows to independently quantify subjects' transition uncertainty, observation uncertainty, and the likelihood of random responding. Since this is the first empirical investigation of this task, we also examined whether the approximate Bayesian learning model suggested by Yu and Dayan [13] accounts better for participants' performance than an optimal Bayesian model.

Interestingly, in a recent meta-analysis, a distinct pattern of results was reported in deterministic and probabilistic contexts: In the former, OCD patients showed non-specific impairments, whereas in the latter preliminary evidence for overly flexible behavior was found [7]. This might suggest that distinct processes govern patients' behavior in accordance with

whether feedback is reliable or noisy [12]. However, hitherto no study has directly compared the two types of tasks. Thus, the secondary goal of this study is to examine whether a different pattern of results emerges for these two types of tasks.

## Results

### Learning task

58 participants recruited from the general population, with a wide range of OC symptoms (~40% participants scored above the clinical cutoff), performed a modified spatial cueing task (see Fig 1), previously used to substantiate an influential model of approximate Bayesian learning in the brain [13]. On each trial, participants were presented with three arrow cues pointing either left or right. Participants were told that one of these cues predicts the location of the

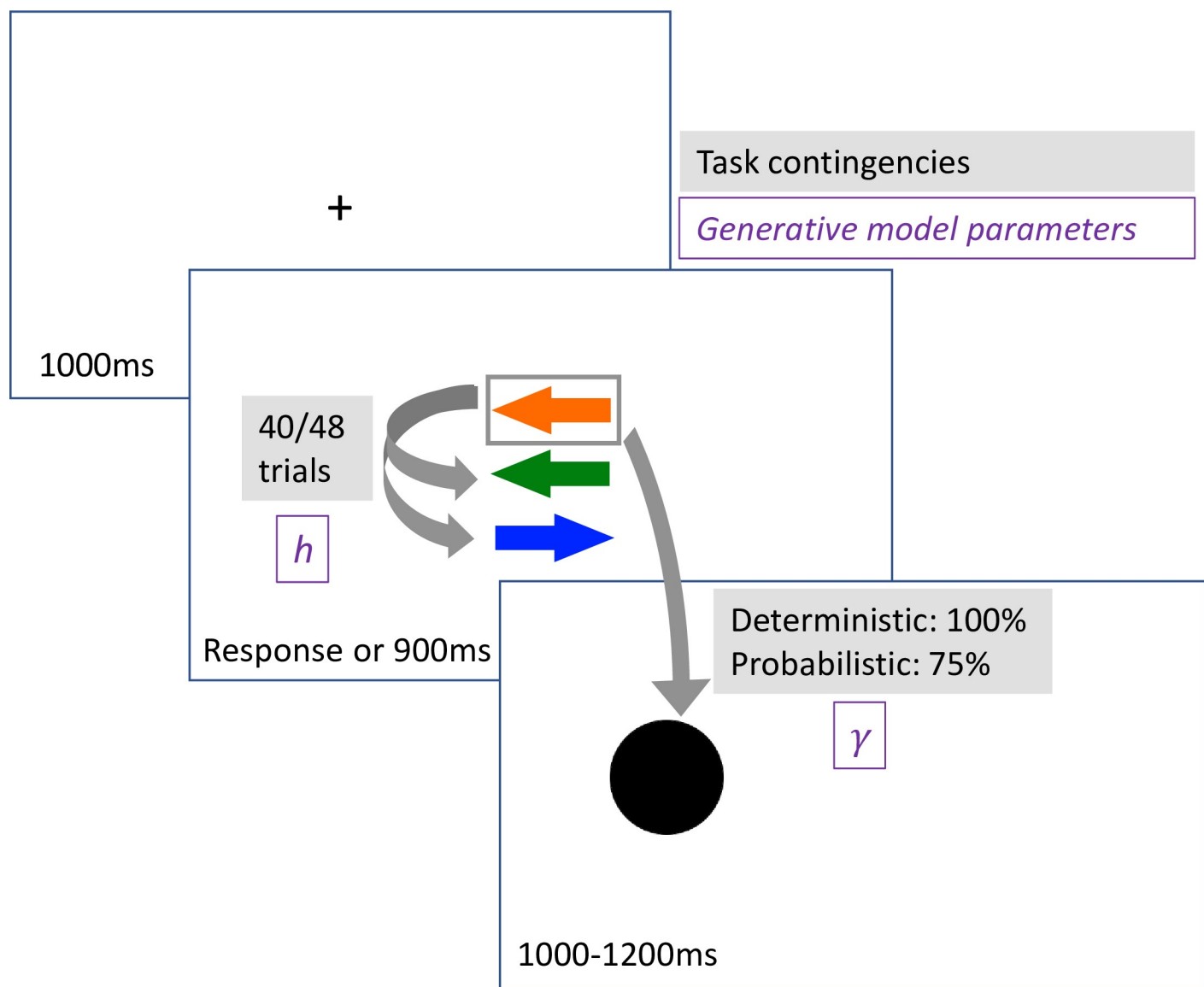

**Fig 1. Illustration of the reversal learning task, and the parameters of the Bayesian generative models used for the computational analysis.** $h$–transition uncertainty; $\gamma$–cue validity (with 1- $\gamma$ representing observation uncertainty).

subsequent target (black circle), and that once in a while a contingency shift (hereafter "shift") will occur–the hitherto predictive cue became irrelevant and a different cue became predictive. Participants' task was to predict the location of the target by pressing the left or right arrow keys (before it appears). The task included two main conditions, a *deterministic condition* where all trials were valid (i.e. the relevant cue always predicted the location of the target), and a *probabilistic condition*, with 75% cue validity. Each condition included 88 trials, with a single shift occurring after either 40 or 48 trials (counterbalanced across participants and conditions).

## Preliminary behavioral analysis

Prior to examining the data through the lens of a theory-based computational model, which is the focus of the current work, we present the basic behavioral findings. As a crude single-trial measure of accuracy, we first examined whether the participant's response matched the orientation of the relevant cue. Overall, participants' mean accuracy in the deterministic condition was 0.94 pre-shift and 0.93 post-shift. Mean accuracy in the probabilistic condition was 0.79 pre-shift and 0.78 post-shift. As depicted in Fig 2, participants reached an asymptote (of ~0.97) quickly in the deterministic condition, with performance dropping immediately after the shift. Conversely, in the probabilistic condition, participants' mean performance was more unstable pre-shift, with decreases likely reflecting incorrectly interpreting probabilistic errors as real contingency shifts. A more stable increase in performance was observed post-shift, likely reflecting a belief that a shift has already occurred, such that accumulated knowledge appeared more reliable.

Next, we used logistic multilevel regressions [16] to examine the effects of OC symptoms as measured by the Obsessive Compulsive Inventory-Revised (OCI-R [17]) on performance. A marginally significant effect was found for OCI-R (total) scores in the probabilistic condition ($\beta$ = -0.009, $Z$ = -1.89, $p$ = .058) but not in the deterministic condition ($\beta$ = -0.008, $Z$ = -1.22, $p$ = .222). A more specific measure can be obtained by focusing on trials in which participants' responses matched only one of the three cues (henceforth *disambiguating trials*). Accuracy in these trials reveals whether the correct cue was chosen. Using this cleaner measure resulted in a significant effect of OCI-R in the probabilistic condition ($\beta$ = -0.015, $Z$ = -2.12, $p$ = .027). Comparing participants' pre- vs. post-shift performance in the probabilistic condition showed that in both types of analyses, high OCI-R scores predicted inferior performance post-shift (all trials: $\beta$ = -0.013, $Z$ = -2.16, $p$ = .031; disambiguating trials: $\beta$ = -0.022, $Z$ = -2.42, $p$ = .015) but not pre-shift (all trials: $\beta$ = -0.005, $Z$ = -0.70, $p$ = .485; disambiguating trials: $\beta$ = -0.007, $Z$ = -0.09, $p$ = .41) although the interaction was not significant ($p$'s $\geq$ .23).

On the surface, these data seem to suggest that high OC participants' inferior performance is due to perseveration, naturally evident only after the shift, thus challenging our hypothesis. However, inspecting participants' choices in disambiguating trials revealed that the proportion of errors that can be attributed to perseverative selection of the cue that was relevant pre-shift did not increase (and in fact was non-significantly lower) for participants with high OCI-R scores ($\beta$ = -0.006, $SE$ = 0.012, $Z$ = -0.47, $p$ = .63). Moreover, inspecting how participants' performance changed within blocks revealed a trend associating higher OCI-R scores with inferior performance at later stages of the pre-shift block (see Fig 3A; trial x OCI-R interaction: $\beta$ = -0.0008, $Z$ = -1.72, $p$ = .085). Together, this pattern might suggest that high OC participants' inferior performance at later stages of the task did not result from perseveration. Rather, it potentially reflects either premature attempts to seek a new relevant cue before the relevant cue actually changed, or a difficulty establishing the new cue. Interestingly, this effect was found only for the probabilistic condition, where the discrimination of real contingency shifts

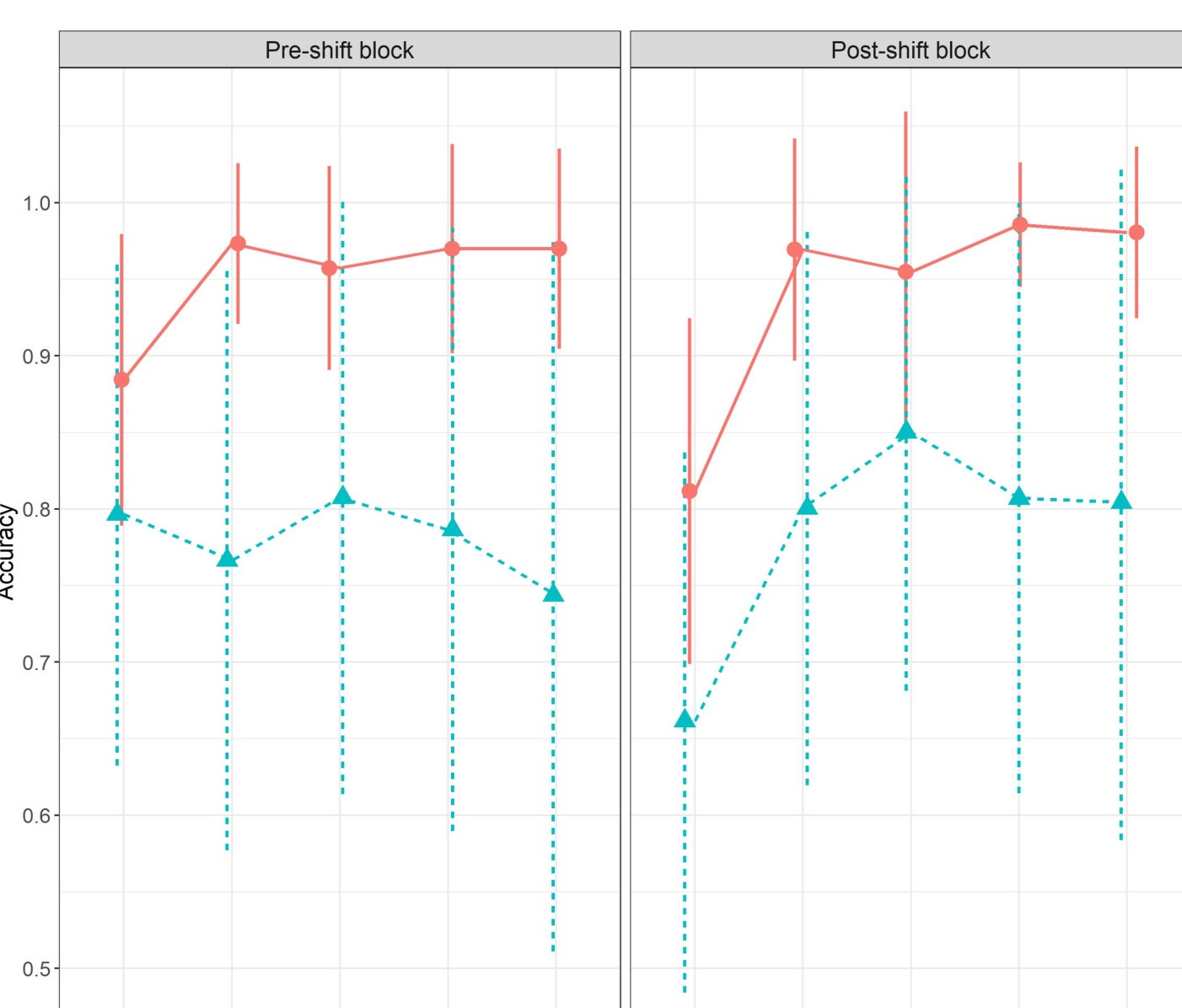

**Fig 2. Mean performance over time pre- and post-shift, in the deterministic and probabilistic conditions.** Error bars represent individual differences (±1SD).

from noise is more challenging. However, accuracy is only a crude measure of the cognitive processes governing participants' behavior. Therefore, we use modeling to determine the processes responsible for this underperformance, and to examine our main hypothesis.

### Bayesian learning computational models

First, we aimed to determine which of two classes of Bayesian learning models best describes participants' behavior: an optimal Bayesian change-point (BCP) model that simultaneously

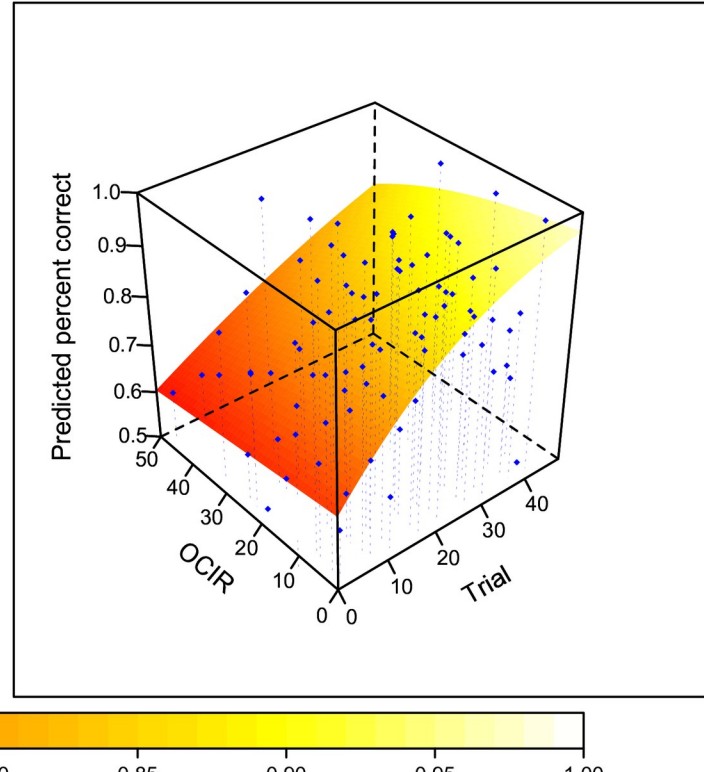

**A)Pre-shift block**

**B)Post-shift block**

**Fig 3. Performance (accuracy) as a function of OCI-R scores, and trial number.** The surface plot depicts the results (predicted scores) of a logistic multilevel regression. Dots represent the actual accuracy, binned in intervals of 5 trials, and 10 percentiles on the OCI-R (percentile binning was used because of the skewed distribution of the OCI-R). The figure shows that higher OCI-R scores correlated with a decrease in performance in the late stages of the pre-shift block (A), as well as in the entire post-shift block (B).

learns about all three arrow cues, or a selective attention (SA) model that focuses on a single arrow on each trial. In both models, transition uncertainty was formalized as a participant's estimate of the probability that the previously accumulated knowledge is no longer relevant, as determined by the free parameter $h$. Estimated cue validity was determined by the parameter $\gamma$, and observation uncertainty was correspondingly defined as $1-\gamma$. The probability for random responding was parameterized by $\varepsilon$ (very low values of this parameter can also be used to indicate random performance, which justifies exclusion). These models were compared with two simple benchmark models where knowledge was not accumulated over trials. This stage is crucial for determining whether participants actually follow a Bayesian model when solving the task.

### Bayesian change-point (BCP) model

The agent tracks the probability of each cue being the relevant cue given all previous observations. Information observed before the last shift is not useful for determining the relevant cue, and thus the agent must infer how long ago the relevant cue has last changed (i.e. *run-length*). The agent estimates the likely run-lengths on a given trial using a Bayesian change-point detection algorithm [18,19]; see Eqs 1–8). This algorithm weights evidence accumulated on previous trials by the probability that a shift did not occur yet (as given by the run-length distribution). So, for example if there is a high estimated probability that a shift occurred on the last trial (*t-*

*1*), evidence that preceded it is disregarded. Conversely, if there is a high probability that a shift occurred *x* trials ago, evidence accumulated during these *x* trials is given a higher weight than evidence accumulated before trial *t-x*. After an estimated shift the distribution over cues is simply the uniform distribution, reflecting the belief that once the previous knowledge is no longer relevant, learning starts anew.

The run-length distribution itself is also updated by integrating: a) evidence for a shift on trial *t* (e.g., a consistent mismatch between the actual location of the target, and its expected location as given by the different cues, each weighted by its estimated probability), and b) the prior probability that a relevant cue on trial *t*-1 is no longer relevant on trial *t* (i.e. transition uncertainty). We examined both a model where this prior probability is assumed to be constant across trials, and a model in which it increases as a function of the run-length (indicating a belief that shifts become more likely over time), following a previously used simple exponential function [20] (see Eq 2).

When learning from feedback, the model takes into account the possibility that (particularly under probabilistic contingencies) the relevant cue does not always point to the right direction. This is reflected by the estimated cue validity parameter ($\gamma$, which is the complement of the observation uncertainty). We make the simplifying assumption that $\gamma$ remains constant across trials, although it is likely learned during the task. This common simplification [19] allows us to use an analytical solution for the recursive update, which facilitates model fitting. Finally, response probabilities on trial t+1 are determined by the orientation of all cues, weighted by their estimated probabilities, and a fixed probability of responding randomly ($\varepsilon$; see Eq 6).

## Selective attention (SA) model

On each trial, the agent focuses on a single cue (rather than learning about all three cues). The agent then decides whether to stick with this cue or not, based on the agent's confidence that this cue is indeed the relevant one ($\lambda$). Yu and Dayan [13] have shown that $\lambda_t$ can be computed recursively as a function of three factors: Prior confidence ($\lambda_{t-1}$); transition uncertainty ($h$), such that greater transition uncertainty implies that the cue is no longer relevant, reducing the relevance of prior confidence in this cue; and estimated cue validity ($\gamma$), which amplifies learning from feedback at trial *t* at the expense of relying on prior confidence. The equations governing this learning process (Eqs 11–16) can be also found in Yu and Dayan [13].

Following each trial, the agent switches attention with probability 1- $\lambda_t$. Whereas in Yu and Dayan [13] this relationship is deterministic (i.e. switches occur when $\lambda_t < .5$), here we assume a probabilistic relationship, to deal with fitting issues described below [19]. Participants are assumed to follow an $\varepsilon$-greedy policy, responding in accordance with the attended cue with a probability of 1-$\varepsilon$, and responding randomly with a probability of $\varepsilon$.

A major obstacle in fitting this model is the fact that the experimenter has no definite knowledge of which cue the participant attends to on a given trial (because participants respond with the right/left keys with no explicit selection of cue). Therefore, we were unable to use the full model suggested by Yu and Dayan (where $\gamma$ is learned over trials) to fit participants' data. However, following the approach suggested by Wilson and Niv [19], we can infer a *distribution* over attended cues given the history of participants' actual responses, observed cues and targets. This is done by applying the change-point algorithm to infer the run-length since the last time the participant switched their attention to a different cue, where the prior probability of such a switch is 1- $\lambda_t$. This distribution is then used to obtain response probabilities (Eq 24). A detailed description of this algorithm can be found in Eqs 17–24.

  

### Win-stay lose-shift (WSLS) model

In this simple benchmark model, the agent is assumed to focus on a single cue on each trial. After feedback is obtained, the agent sticks with this cue in case of an expected outcome (i.e. when the target's location matches the orientation of this cue) with probability $p_{stay}$, and switches to a different cue in the case of an unexpected outcome with probability $p_{shift}$. It is a simple selective attention model that does not require complex Bayesian learning. Since this model shares that same problem of estimating participant's attended cue as the SA model above, a similar solution was used [19].

### No learning model

This model was designed as an even simpler baseline for examining the absolute fit of the learning models. Here, response probabilities were based only on the proportion of arrow cues pointing at a specific direction, with no learning.

### Model comparison

Model parameters were estimated in a hierarchical Bayesian framework that regularizes individual participants' parameters using group-level parameter distributions, and which typically produces more reliable estimates [21]. Models were compared by using the Widely Applicable Information Criterion (WAIC), and an approximation of the leave-one-out validation (PSIS-LOO), which are state-of-the-art measures of out-of-sample predictive accuracy of Bayesian models [22]. To support the interpretation of these results, these values were used to approximate the *relative* likelihood of each model being the best model by calculating models' weights (Akaike weights for the WAIC, and pseudo-BMA weights for the PSIS-LOO [23]). In addition, we examined each model's absolute fit by using the entire posterior distribution of participants' parameters to generate a distribution of simulated responses (per-participant), and calculating the average match between the participant's actual data and these simulated responses.

As depicted in Fig 4A and 4B, The BCP models had a better fit (lower WAIC and PSIS-LOO values) than both the SA models, and the WSLS models in both conditions. BCP models with constant $h$ (transition uncertainty) were equivalent to changing-$h$ models in the probabilistic condition, but outperformed them in the deterministic condition. Surprisingly, in the deterministic condition, models allowing cue validity ($\gamma$; equal to 1 by definition) to be free performed better. Nonetheless, estimated $\gamma$ in that condition was close to 1 for most participants (inter-quartile range = 0.9904–0.9982). Together, these results led us to focus on the BCP models with constant $h$ and free $\gamma$ for the analyses below. Results of the changing-$h$ model were similar and are reported in the Supporting information (S1 Table).

For most participants, the best-fitting models performed better than chance (0.5), and better than a no-learning model (see Fig 4C and 4D). The prediction of the responses of one participant (colored in red) was close to chance-level in both conditions. The fitted values of $\epsilon$ for this participant were also high (e.g., 0.65 in the probabilistic conditions), implying random responding. This participant was excluded from all analyses, although this exclusion did not significantly alter the results.

### Transition uncertainty and OC symptoms

To obtain a point estimate of the computational parameters of interest, the medians of participants' posterior distributions were used. In accordance with our main hypothesis, OCI-R scores were positively correlated (using a non-parametric permutation test due to the violation

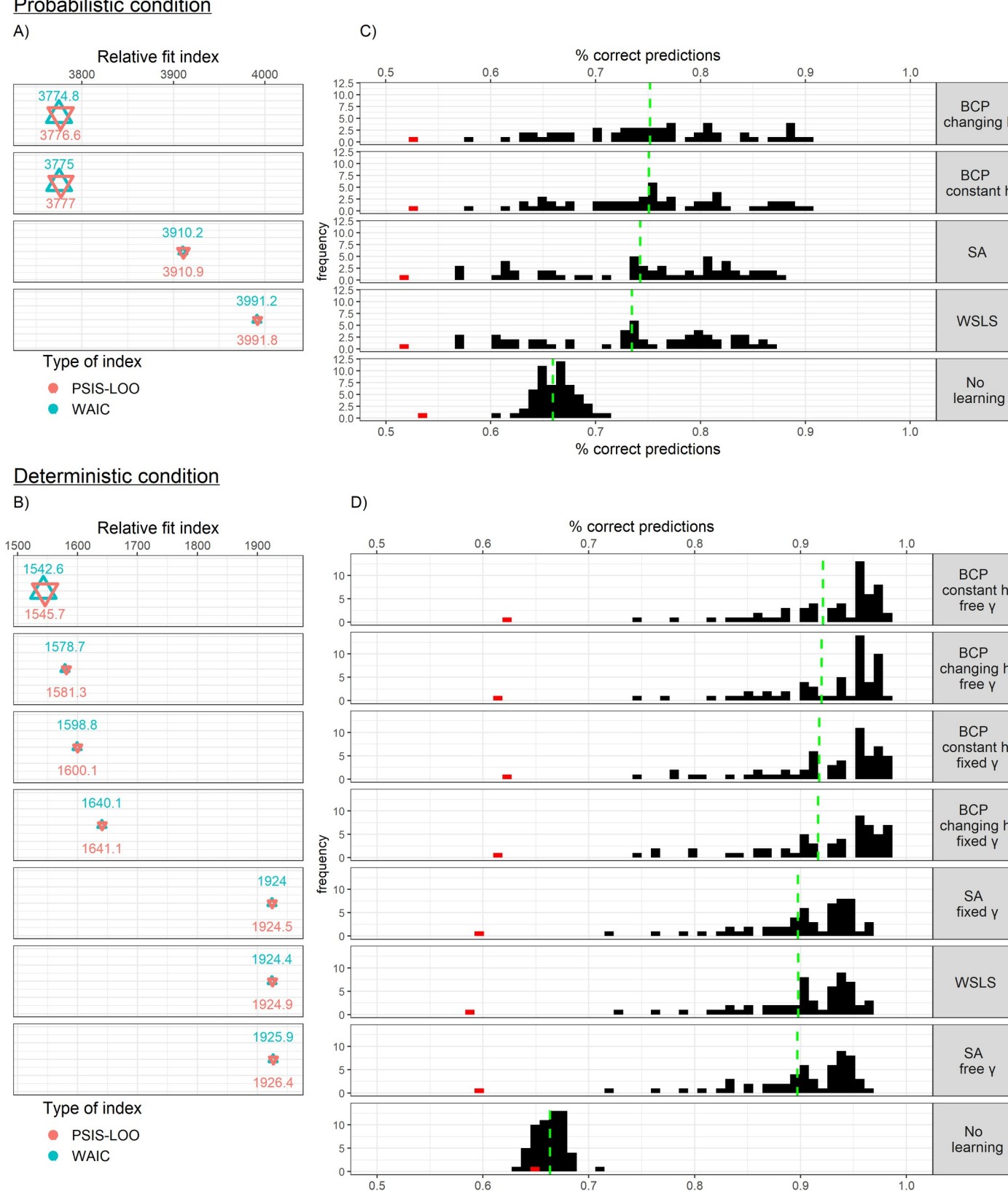

**Fig 4. Model comparison results for the most competitive models (less competitive variations of these models are presented in S2 Table and S2 Text in the Supporting information).** Panels A and B present the relative fit indices (the widely applicable information criteria; WAIC, and an approximation of the leave-one-out validation; PSIS-LOO), with lower values representing better fit. The size of the triangles represent the relative weights (i.e. approximation of the relative likelihood) of the different models. Panels C and D present the distributions (over participants) of the absolute fit, computed as the proportion of correct predictions (i.e. match between model-based simulated responses and actual responses) for each model. The green, vertical, dashed line represents the average absolute fit. The red (outlying) bar represents a participant excluded from all analyses due to this and additional evidence for negligent, chance-level performance. BCP–Bayesian change point model; SA–selective attention model; h–a free parameter determining transition uncertainty; γ–a free (or fixed at 1, in some models) parameter determining the complement of observation uncertainty.

of normality) with transition uncertainty ($h$) in the probabilistic condition ($r = .31$, $p = .017$; Fig 5A), whereas the effect in the deterministic condition was only marginally significant ($r = .24$, $p = .062$; Fig 5B). OCI-R scores did not correlate with observation uncertainty (1-γ) in the probabilistic ($r = .15$, $p = .259$) or deterministic ($r = -.06$, $p = .704$) conditions. Likewise, OCI-R scores also did not correlate with random responding (ε) in the probabilistic ($r = .05$, $p = .732$) or deterministic (r = .06, p = .639) conditions. These results show that underperformance related with OC symptoms is indeed the result of under-weighing accumulated knowledge, and not of perseveration or non-specific stochasticity in the response process.

## OC symptoms and sensitivity to feedback

As outlined above, transition uncertainty and observation uncertainty interact in determining the weight given to feedback (i.e. prediction errors). Thus, this computational setup allowed us to use participants' best fitted parameter values to estimate two trial-level measures of the processing of feedback in the probabilistic condition. First, we examined how unexpected each outcome was to participants (using a measure of surprise; see Eq 9). For example, the case in which all three cues point to one direction but the target appears at the other direction is highly unexpected. Second, we examined the degree to which each outcome made the participants change their beliefs about the relevant cue (using the KL-divergence; see Eq 10). Crucially, not all unexpected feedback leads to learning. Indeed, in the example given above, the unexpected target provides no new information regarding the relevant cue. More generally, high transition uncertainty increases both measures, whereas observation uncertainty increases surprise but decreases model updating. At the extreme case wherein γ = .5, feedback is always unpredictable, yet is completely uninformative about the relevant cue. Thus, examining the feedback processing measures can help better understand the *interaction* between the two uncertainty parameters in high OC participants. Furthermore, these two, partially dissociated types of prediction error [24] have different neural markers [24–28].

Higher OCI-R scores predicted higher surprise in valid trials (β = 0.001, $t = 2.10$, $p = .040$) and lower surprise in invalid trials (β = -0.002, $t = -2.07$, $p = .043$; the interaction was significant: $p = .031$), suggesting that transition uncertainty decreased high OC participants' confidence in their predictions. Indeed, OCI-R scores were positively correlated with trial-level uncertainty (i.e. entropy) regarding the target's predicted location (β = 0.0004, $t = 2.09$, $p = .042$).

In contrast, OCI-R scores were not correlated with model updating in valid (β = 0.00036, $t = 1.86$, $p = .067$) or invalid trials (β = 0.00035, $t = 1.23$, $p = .22$). This might reflect the fact that although OCI-R was not significantly correlated with observation uncertainty, the direction of this relationship was positive (see S1 Table). Recall that transition and observation uncertainty impact model updating in opposite directions, and thus even slightly elevated observation uncertainty may have counteracted the effect of high transition uncertainty on model updating.

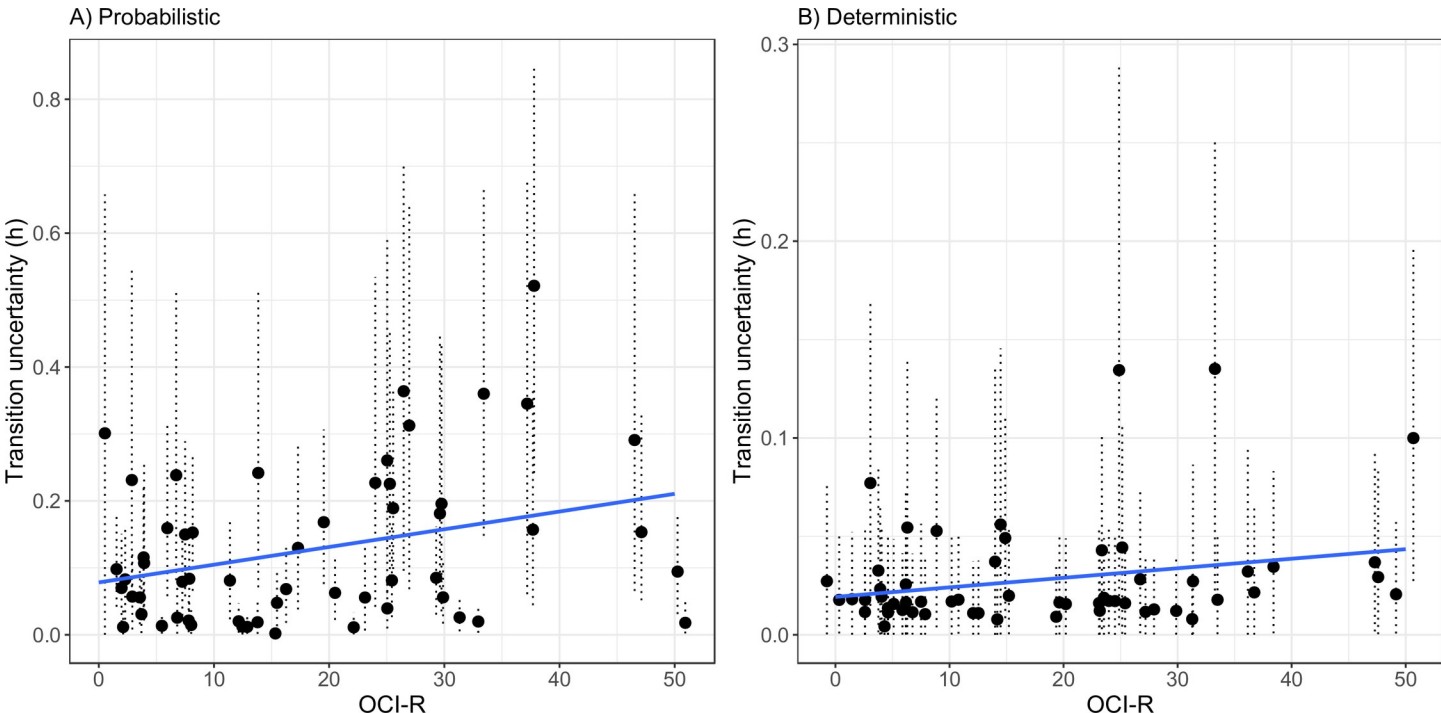

**Fig 5.** Scatterplots depicting the association between OCI-R scores and transition uncertainty fitted values (medians and 95% Bayesian high density intervals), for the probabilistic (A) and deterministic (B) conditions.

## Specificity to OC symptoms

Finally, we sought to examine whether transition uncertainty is related specifically to OC symptoms, or whether this relationship can be accounted for by general distress, anxiety or depression. Transition uncertainty was not significantly correlated with anxious arousal (probabilistic: $r = .12$, $p = .378$, deterministic: $r = .06$, $p = .643$), depressive symptoms (probabilistic: $r = .22$, $p = .10$; deterministic: $r = .20$, $p = .140$) or stress (probabilistic: $r = -.02$, $p = .894$, deterministic: $r = .16$, $p = .238$). Nonetheless, the effect size for depressive symptoms was close to that of OCI-R scores. Examining partial correlations showed that the effect of OCI-R controlling for depressive symptoms was no longer significant ($r = .24$, $p = .075$), although it was stronger than the effect of depressive symptoms controlling for OCI-R scores ($r = .07$, $p = .608$). Thus, whereas OC symptoms seem to play a larger role here, evidence for specificity is limited, and examination with larger studies is required.

## Discussion

The current paper examines the hypothesis that OC symptoms are related with excessive transition uncertainty: an impaired ability to rely on past states when estimating the present and predicting the future [12]. Supporting this hypothesis, participants with high OC symptoms exhibited a tendency to distrust what they have learned in previous trials, rendering them constantly uncertain, indecisive and exploratory. Increased transition uncertainty can explain excessive information gathering (e.g., checking, reassurance seeking) in OCD [1–5] as the reasonable (Bayes-optimal) thing to do when previous knowledge is discounted [14].

These results challenge the common preconception that OCD is characterized by inflexibility [8,29]. A previous meta-analysis showed that there is no robust evidence for a specific

flexibility impairment in OCD [7]. The use of computational modeling in the current study allowed for a more specific and somewhat counterintuitive conclusion–rather than inflexibility, OC symptoms correlated with 'over-flexibility' [30,31], especially under probabilistic contingencies. In contrast to that meta-analysis, no robust underperformance was found under deterministic contingencies. This is likely related to the fact that contingency shifts are easier to detect when feedback is deterministic.

It is important to note that these results do not imply that OCD patients (or individuals with high OC symptoms) necessarily have an explicit belief that the environment is unstable. Indeed, in a recent study only patients' behavior, but not their meta-cognitive beliefs, reflected increased reliance on most recent outcomes [5]. Furthermore, increased reliance on recent outcomes might also result, for instance, from poor memory recall or a lack of confidence in memory. Whereas evidence for poor recall in OCD is scarce [32], distrust in memory has been relatively robust [6,32–34]. However, transition uncertainty might be a mechanism that leads to distrust in memory in OCD: if past states are irrelevant, then predictions based on memory should be regarded as unreliable. Further research is needed to determine to what degree transition uncertainty and memory distrust overlap.

Another important consideration concerns the distinction between heightened (transition) uncertainty and an excessive need to resolve uncertainty (i.e. intolerance of uncertainty [35]), as both can give rise to excessive information gathering [35–37]. Intolerance of uncertainty seems to be supported by a recent study that linked OC symptoms and anxiety to increased information seeking even in a task in which this information had no effect on actual control or performance [37]. Future studies should attempt to dissociate the relative contribution of these interacting processes to patients' performance and symptoms.

Using a BCP model to estimate participants' internal responses to feedback (prediction errors) indicated that OC symptoms made expected feedback more surprising, and unexpected feedback less surprising (see also [38,39]). However, OC symptoms did not correlate with a measure of model updating, suggesting that exploration in this case does not involve over-learning from feedback. Recently, these measures of prediction error were associated with two different electrophysiological subcomponents–the P3a with the surprisal and the P3b with model updating [25]. Consistent with our results, in two studies, only the P3a was increased in OCD [40,41]. OCD research integrating computational modeling with these direct measures of surprisal and updating is required.

Excessive transition uncertainty is expected to affect not only reliance on the past, but also goal-directed behavior. Specifically, if the past and present cannot predict the future, predicting and planning the future consequences of behavior becomes very complicated [12]. Prominent models of OCD focus on impairments in goal-directed control and overreliance on habits [42]. An interesting question for future research is whether these impairments in goal-directed control are the result of increased transition uncertainty. Indeed, previous theories have suggested that when the consequences of goal-directed strategies are unpredictable, compensatory, habitual behavior is likely to emerge [12,43]. Notably, for habits to emerge, an opportunity to learn habits (i.e., over trained S-R mapping) is necessary. One possibility is that no perseveration was found in the current study because habit learning is relatively unlikely in the current task, which includes many possible S-R combinations (i.e. 8 combinations of arrows X 2 responses) to be learned over less than 50 trials–leaving an insufficient amount of training per contingency.

## Methodological implications

Despite the widespread theoretical influence of the model suggested in Yu and Dayan [13], the current study is the first to empirically examine the modified spatial cueing task used to

instantiate this model. This task differs from classic reversal learning tasks in several ways. As delineated below, these differences could make the task and the models developed here useful for research in other contexts.

First, objective feedback does not depend on participants' behavior, allowing participants to concurrently learn about all cues. Indeed, in contrast to the selective attention model originally proposed by Yu and Dayan [13], our results suggest that participants distribute attention across all cues. This was the case under both probabilistic and deterministic contingencies. However, attentional constrains are expected to have a greater role as the number of cues increase (e.g., 5 cues instead of 3). In addition, this characteristic can be important when one is interested in focusing specifically on uncertainty regarding action-independent transitions (i.e. predictions regarding the evolution of states). This can be contrasted with uncertainty regarding action-dependent transitions (i.e. predictions regarding the consequences of one's actions), which is likely to play a larger role in tasks in which feedback depends on behavior [12].

Second, whereas we developed an alternative-forced-choice version of the task, examining participants predictions, the original paradigm was designed to capture the attentional processes involved in responding to a cued target (see also [44]). The models developed here can be readily used (when combined with a response model appropriate for the prediction of response times) to investigate the Bayesian processes involved in spatial cueing under uncertainty (see S1 Text).

## Conclusions and future directions

The current study has shown that high OC symptoms are related with a reduced reliance on past knowledge, which can explain the OC phenomenology of excessive uncertainty and doubt, and the ensuing need to repeatedly verify what should have been already known [12]. This stands in stark contrast to the idea that OCD is characterized by inflexible, perseverative behavior, corresponding with over-reliance on past knowledge.

It is important to replicate these findings in additional non-clinical and clinical samples. Relatedly, the current study was underpowered to robustly examine the specificity of the effects for OC symptoms (vs. general anxiety or depression). Future studies would benefit from using a large sample allowing to better characterize the contribution of transition uncertainty to different types of symptoms by extracting independent dimensions of psychopathology (e.g., using factor analysis with multiple scales; see [42]).

It should also be noted that the results of this study depend on the validity of the chosen BCP model, and on the assumption that people use some sort of Bayesian inference in this task. We examined several different Bayesian and non-Bayesian computational models. Nevertheless, other models can of course be developed. The behavioral pattern of increased reliance on more recent feedback can obtain a different theoretical meaning in computational models that make different assumptions (e.g., a reinforcement learning model with a 'forgetting' parameter [45]).

Finally, the next step is to develop a more ecological design, examining the role transition uncertainty plays in clinically relevant contexts. This requires the addition of the real-life factors that likely play a moderating role in OCD–such as inducing a potential for harm, using patient-tailored anxiogenic stimuli, manipulating motivations, etc. This has the potential to allow for the development of ecological, individualized computational models of real clinical symptoms, potentially leading to the development of novel, personal interventions.

## Methods and materials

### Participants

We recruited 58 participants from the general population. The use of non-patient samples for OCD research is common and recommended [46], and has the advantage of allowing to

measure the specificity of the results to OC symptoms (vs. non-specific anxiety or depression) using the same sample. One outlying participant was excluded due to strong evidence for random responding (see Fig 4). The final sample included 36 (63.18%) women, and participants were on average 24.21 years old (*SD* = 3.05) with 13.88 years of education (*SD* = 1.47). All participants had normal or corrected to normal vision.

### Ethics statement

The study was approved by the research ethics committee of the social science faculty of the Hebrew University of Jerusalem. All participants provided written informed consent prior to participation.

### Reversal learning task

On each trial, participants were presented with three arrow cues pointing either left or right. Participants were told that one of these cues predicts the location of the subsequent target (black circle). Participants' task was to predict the location of the target by pressing the left or right arrow keys. The target appeared immediately after a response, or after 900ms at the absence of a response. Participants had to learn from experience which cue predicts the target's location.

The task included two main conditions, a *deterministic condition* where all trials were valid (i.e. the relevant cue always predicted the location of the target), and a *probabilistic condition*, with 75% cue validity. In the latter condition, participants were told that the arrow will predict the location of the target in most but not all trials, but the exact rate (i.e., cue validity) had to be estimated. After a random number of either 40 or 48 trials (counterbalanced, such that for half the participants the probabilistic condition included 40 trials before the shift and the deterministic condition included 48 trials before the shift, whereas for the other half of participants this was flipped), a contingency shift occurred–the hitherto predictive cue became irrelevant and a *different* cue became predictive. Participants were explicitly told that the relevant cue will change at some point. Participants first performed a short deterministic training run (consisting of 32 trials before the shift and 10 trials after). Then, participants performed two blocks of trials, one deterministic and then one probabilistic, each including 88 trials. We fixed the order of the conditions because a pilot study indicated that the probabilistic condition is difficult to understand without proper experience with a simpler, deterministic, block.

The task included a second part (an attentional cueing task), in which participants were asked to press the space bar when they detected the target. Participants' response times in this part were intended as an additional measure of the processing of feedback. Whereas these additional results (see S1 Text in the Supporting information section) were consistent with the results reported here, they should be taken with caution because participants' response times were only weakly related to the orientation of the relevant cue. The data and code for the computational models can be found in: http://doi.org/10.17605/OSF.IO/D6B3M. The full study protocol can be found also in: https://doi.org/10.17504/protocols.io.97nh9me

### Bayesian learning computational models

As described above, we compared the performance of two classes of Bayesian learning models: an optimal Bayesian change-point (BCP) model that simultaneously learns about all three arrow cues, or a selective attention (SA) model that focuses on a single arrow on each trial.

## BCP model

In this optimal observer model, the agent tracks the probability of each cue being the relevant cue given all previous observations ($p(c|D_{1:t})$, where $c \in \{top, middle, bottom\}$). Because the task includes unsignaled shifts (of the relevant cue), and information observed before the last shift is not useful for determining the relevant cue, the agent must infer how long ago the relevant cue has last changed (i.e. *run-length*). Following the approach of Wilson and Niv [19], we used the Bayesian change-point algorithm developed by Adams and McKay [18], where participants track the run-length distribution ($p(l_t)$). The run length increases by one following each trial and resets to zero at each change-point. Since change points are only probabilistically known, the likely run length at each trial is represented by a (categorical) distribution. Then, prior to responding (on trial t+1), the agent must integrate previous experience (regarding the relevant cue) accounting for the probability that this experience is still/no longer relevant (i.e., in case of a change-point):

$$p(c|D_{1:t}) = \sum_{l_{t+1}} p(c|l_{t+1}, D_{1:t}) \sum_{l_t} p(l_{t+1}|l_t, D_{1:t}) p(l_t|D_{1:t}) \tag{1}$$

where $p(l_{t+1}|l_t, D_{1:t})$ reflects the prior probability that the relevant cue changes on trial t+1. We examined both a model where this prior probability is assumed to be constant across trials (i.e., $p(l_{t+1} = 0|l_t, D_{1:t}) = h$), and a model where it increases as a function of the run-length, following a previously used simple exponential function [20]:

$$p(l_{t+1} = 0|l_t, D_{1:t}) = 1 - e^{(-hl_t)} \tag{2}$$

The distribution over cues following a switch ($p(c|l_{t+1} = 0, D_{1:t})$) is simply a discrete uniform distribution. Note that whereas in the model a switch is followed by a uniform distribution over all three cues, in the task the same cue is never resampled after a switch. The reason for defining $h$ this way is that we focused on the uncertainty regarding state transitions rather than the probability for a change in contingencies (i.e. volatility). Thus, for example, $h = 1$ corresponds with completely discounting previous knowledge (whereas in a completely volatile environment previous knowledge can be used to infer which cue is *irrelevant*).

When the relevant cue persists, this distribution is estimated recursively, by integrating the previous estimate with the current outcome ($S_t$):

$$p(c|l_{t+1} = l_t + 1, D_{1:t}) \propto p(S_t|c) p(c|l_t, D_{1:t-1}) \tag{3}$$

where the relationship between outcome and relevant cue reflects cue validity (free parameter $\gamma$):

$$p(S_t|c) \equiv \begin{cases} \gamma & if\ S_t = (c) \\ 1 - \gamma & if\ S_t \neq (c) \end{cases} \tag{4}$$

where $(c)$ represents the direction to which each arrow cue points.

For a complete model, one must recursively update also the run-length distribution, which is given by:

$$p(l_t|D_{1:t}) \propto \sum_c p(S_t|c) p(c|l_t, D_{1:t-1}) \sum_{l_{t-1}} p(l_t|l_{t-1}, D_{1:t-1}) p(l_{t-1}|D_{1:t-1}) \tag{5}$$

where the first term on the right side of Eq 5 is obtained by marginalizing over Eq 3, and the second term is similar to $p(l_{t+1}|l_t, D_{1:t})$ above.

Finally, in the response model reported above, response probabilities on trial t+1 (defined as $P(R_{t+1})$), *where* $R \in \{left, right\}$) were determined by the probability distribution over $c$ and a fixed probability of responding randomly (free parameter $\epsilon$):

$$p(R_{t+1}) = (1 - \epsilon) \sum_c p(R_{t+1}|c)p(c|D_{1:t}) + 0.5\epsilon \qquad (6)$$

where $p(R_{t+1}|c)$ is simply an identity matrix mapping right-key responses to right-pointing arrows.

We examined two additional response model. First, a *matching* response model where cue validity (γ) also influenced the response probability. In this model, response probability was assumed to track the probability of the target appearing at specific location:

$$p(R_{t+1}) = (1 - \epsilon) \sum_c p(S_{t+1} = (c)|c)p(c|D_{1:t}) + 0.5\epsilon \qquad (7)$$

To illustrate, in such a matching response model, when $\gamma = 0.5$ (i.e. maximal observation uncertainty, where the target location is assumed to be unrelated to any of the cues) participants will always respond randomly.

Second, we sought to examine a maximizing response model, where despite learning about all three cues, the agent responds only in accordance with the most likely cue (rather than averaging across cues). However, introducing an argmax statement impeded the convergence of the model, most likely because such terms often obstruct the smoothness of the posterior. Thus, we took a different approach by introducing an additional 'inverse temperature' parameter β, which controlled the overweighing of the most likely cue in a continuous manner. Specifically, Eq 6 was replaced with:

$$p(R_{t+1}) = (1 - \epsilon) \sum_c p(R_{t+1}|c) \frac{p(c|D_{1:t})^\beta}{\sum_c p(c|D_{1:t})^\beta} + 0.5\epsilon \qquad (8)$$

Thus, higher β values result in a more maximizing response style, where values close to 1 indicate no overweighting of the most likely cue. Importantly, $\beta$ had a lower bound at 1, because we did not want this additional parameter to control random or 'no-learning' responding (which was already accounted for by the other parameters). For brevity, we report only models with the first response model (Eq 6) in Fig 4 above, whereas the performance of these two alternative response models is reported in the Supporting information section (S2 Table and S2 Text).

Finally, information-theoretic measures of feedback processing were calculated as follows. First, surprisal indicates how unpredictable the outcome ($S_t$) was, and is calculated as:

$$I_t = -log \left[ \sum_c p(S_t|c)p(c|D_{t-1}) \right] \qquad (9)$$

The second measure, KL divergence, indicates the degree to which the outcome made the participant change their beliefs about the relevant cue:

$$KL_t = \sum_c (c|D_{1:t}) log \left[ \frac{p(c|D_{1:t})}{p(c|D_{1:t-1})} \right] \qquad (10)$$

Trial-level uncertainty (entropy) was calculated as the expectation of Eq 9.

## SA model

This model follows the original model suggested by Yu and Dayan [13] with several modifications. On each trial, the agent focuses on a single cue, denoted by $c_t^*$. The agent then decides whether to stick with this cue or switch to another cue, based on its confidence that the current cue is indeed the relevant one, defined as $\lambda$. Following each trial, the agent is assumed to switch attention with a probability $1-\lambda_t$.

After observing the outcome on trial $t$ (denoted by $S_t$), the agent computes the probability that the currently attended cue was in-fact the relevant cue as:

$$\lambda_t \equiv p(c_t^*|D_t) = \frac{p(c_t^*, S_t|D_{t-1})}{p(c_t^*, S_t|D_{t-1}) + p(\neg c_t^*, S_t|D_{t-1})} \tag{11}$$

Eq 11 comprises the joint probability of observing the outcome while the attended cue is correct and that of observing the outcome while the attended cue is incorrect. The former joint probability (brackets in Eq 12) considers two events: either this cue was correct on $t$-1 and no shift has occurred, or a different cue was correct on $t$-1, but a shift has occurred (and now the attended cue became relevant, the probability of which is equal to $0.5h$):

$$p(c_t^*, S_t|D_{t-1}) = p(S_t|c_t^*)[(1-h)\lambda_{t-1} + 0.5h(1-\lambda_{t-1})] \tag{12}$$

while:

$$p(S_t|c_t^*) = \begin{cases} \gamma & \text{if } S_t = (c_t^*) \\ 1-\gamma & \text{if } S_t \neq (c_t^*) \end{cases} \tag{13}$$

The latter joint probability can be approximated by:

$$p(\neg c_t^*, S_t|D_{t-1}) \approx 0.5[h\lambda_{t-1} + (1-h)(1-\lambda_{t-1})] \tag{14}$$

where 0.5 reflects the fact that an irrelevant cue has a 50% chance of predicting the target's location. The term in the brackets considers two events: either this cue was correct on $t$-1 but a shift has occurred, or this cue was incorrect on $t$-1 and a shift did not occur.

Finally, on the first trial of each new context (i.e. on $t = 1$ and after a switch), before observing feedback, the agent's prior confidence in the attended cue is given by the parameter $\lambda_0$, and Eqs 12 and 14 are replaced with:

$$p(c_t^*, S_t|D_{t-1}) = p(S_t|c_t^*)\lambda_0 \tag{15}$$

and:

$$p(\neg c_t^*, S_t|D_{t-1}) = 0.5(1-\lambda_0) \tag{16}$$

Whereas in the models reported above (Fig 4) $\lambda_0$ was a free parameter, we also examined the fit of SA models in which $\lambda_0$ was fixed at 0.5. This corresponds with the original model of Yu and Dayan [13], in which when $\lambda_0 < 0.5$ the agent switches its attention to a different cue. Thus, to start attending to a cue (even if arbitrarily), the agent must believe that this cue is at least as likely to be correct as it is to be incorrect. For brevity, we report only models with a free $\lambda_0$ in the results section because these models consistently had a better fit (see S2 Table, models 5–6 vs. models 10–12).

A major obstacle for fitting this model (as well as the model of Yu and Dayan [13]) to data is the fact that we have no definite way of knowing which cue the participant attends to on a given trial (because participants respond with the right/left keys with no explicit selection of cue). To overcome this issue we followed the approach suggested in Wilson and Niv [19], in

which the agent's attended cue is estimated within the model. That is, although we cannot be confident that the agent attended to a specific cue on trial $t$, we can use the history of cues and targets presented to the agent, as well as the history of the agent's actual responses to estimate the *probability* that the agent attended to this cue. Moreover, the learning model presented above (Eqs 11–16) provides us with a probability for an attentional switch at trial $t$ (which is 1-λt), which we can use as the prior probability for a change-point in the distribution over attended cues.

Specifically, we define $p(c^A|D_{1:t})$ as the distribution over the participant's potential foci of attention after observing feedback on trial $t$. That is, instead of reflecting the distribution over cues from the agent's perspective (as in the BCP model above) we now model the distribution over the agent's attended cue. We use a modified change-point algorithm, where $p(l_t^A|D_{1:t})$ is a distribution that reflects our posterior estimate of the number of trials (i.e run-length) since the last attentional switch (to recap: in this model, change-points reflect the agent's switches in attention, rather than the dynamics of the task, as in the BCP model above). The distribution of $c^A$ is then given by:

$$p(c^A|D_{1:t}) = \sum_{l_t^A} p(c^A|l_t^A, D_{1:t})p(l_t^A|D_{1:t}) \tag{17}$$

The first part of Eq 17 can be computed recursively via:

$$p(c^A|l_t^A, D_{1:t}) \propto p(R_t|c^A)p(c^A|l_t^A, D_{1:t-1}) \tag{18}$$

Note that Eq 18 is parallel to Eq 3, with the exception that the likelihood now corresponds with the participant's response on trial $t$ (rather than the feedback on trial $t$). Stated otherwise, the participant's response on trial $t$ is treated as data in the model inferring the cue the participant has most likely attended to on that trial. Thus, the first part of Eq 18 is computed in accordance with the respective response model. In the simple response model (used in Fig 4) it is equal to:

$$p(R_t|c^A) = \begin{cases} (1-\epsilon) + 0.5\epsilon & \text{if } R_t = (c^A) \\ 0.5\epsilon & \text{if } R_t \neq (c^A) \end{cases} \tag{19}$$

Whereas in the case of a matching response model (not reported in the results section, due to inferior fit; see S2 Table, model 5 vs. model 6) it is equal to:

$$p(R_t|c^A) = \begin{cases} (1-\epsilon)\gamma + 0.5\epsilon & \text{if } R_t = (c^A) \\ (1-\epsilon)(1-\gamma) + 0.5\epsilon & \text{if } R_t \neq (c^A) \end{cases} \tag{20}$$

The second part of Eq 18 is given by:

$$p(c^A|l_t^A, D_{1:t-1}) = \begin{cases} \dfrac{1}{3} & \text{if } l_t^A = 0 \\ p(c^A|l_{t-1}^A, D_{1:t-1}) & \text{otherwise} \end{cases} \tag{21}$$

where 3 is the number of arrows (corresponding with a uniform distribution).

The second part of Eq 17 is also computed recursively via:

$$p(l_t^A|D_{1:t}) \propto \sum_{c_t^A} p(c^A|l_t^A, D_{1:t}) \sum_{l_{t-1}^A} p(l_t^A|l_{t-1}^A, D_{1:t-1})p(l_{t-1}^A|D_{1:t-1}) \tag{22}$$

, where

$$p(l_t^A|l_{t-1}^A, D_{1:t-1}) = \sum_{c_{t-1}^A} p(switch_t|l_{t-1}^A, c_{t-1}^A)p(c_{t-1}^A|l_{t-1}^A) \tag{23}$$

Eq 23 corresponds to the (experimenter's) estimate of the probability that the agent has switched attention on trial $t$ (resulting in a $l_t^A = 0$). It shows that the probability with which the agent (from its own perspective) switches ($p(switch_t) = 1-\lambda_t$) is in fact computed for each possible attended cue ($c_{t-1}^A$, given by Eq 17) and possible run-length ($l_{t-1}^A$).

Whereas Eqs 19 and 20 defined the response probabilities required for the estimation of the likelihood of the *actual* response at trial $t$ given an attended cue (to infer the attended cue), they are also used to predict the participant's response on trial $t+1$. This requires the computation of the marginal probability for a specific response. Thus, the probability that the participant responds with the *right arrow key* at trial $t+1$ is given by:

$$p(R_{t+1}=`right') = \sum_{c^A} p(R_{t+1}=`right'|c^A)p(c^A) \tag{24}$$

where $p(c^A)$ is derived from Eq 17.

Finally, note that in this model, there is a slight inconsistency in the interpretation of contingency shifts between the agent's learning model, and the (experimenter-level) model used to estimate the agent's attended cue. That is, in the agent's learning model, $h$ corresponds with the probability that a different cue is now relevant. Thus, for example, if the agent was completely certain of a cue on trial $t-1$ ($\lambda_{t-1} = 1$), but estimates that $h = 1$, their confidence in this cue at trial $t$ (derived from Eq 11) becomes zero. In contrast, Eq 21 shows that in the case of an attentional switch, the agent transfers to a uniform distribution over the three cues (including the cue attention was just switched from). This inconsistency is inherent to Yu and Dayan's [13] original model, where switches of attention lead to exploration in which all cues are equally likely to be sampled. Because we wanted to preserve the original model proposed by Yu and Dayan's [13] we used their learning equations despite the resulting inconsistency. However, we also examined the sensitivity of our results to using an alternative learning model, where all three cues are assumed to be equally probable following shifts (i.e. a uniform distribution). This required a slight change in Eqs 12 and 14. Particularly, the joint probability of observing the outcome while the attended cue is correct becomes:

$$p(c_t^*, S_t|D_{t-1}) = p(S_t|c^*)\left[\left(1 - h + \frac{h}{3}\right)\lambda_{t-1} + \frac{h}{3}(1 - \lambda_{t-1})\right] \tag{25}$$

Where the first h/3 reflects the probability that a shift has occurred but then the attended cue became relevant again, and the second $h$/3 has replaced the previous 0.5$h$, due to the assumption that in this model, the probability for a shift from a different cue to the attended cue is only one third.

The joint probability of observing the outcome while the attended cue is incorrect becomes:

$$p(\neg c_t^*, S_t|D_{t-1}) \approx 0.5\left[\frac{2}{3}h\lambda_{t-1} + (1 - h)(1 - \lambda_{t-1})\right] \tag{26}$$

Where 2/3 comes from the idea that if the attended cue was correct on trial $t-1$ and a shift has occurred, there is still a 1/3 chance that the attended cue will become relevant again. Critically, these modifications did not alter the model comparison results (for example in the best fitting SA model–model 5 in S2 Table–this change led to a WAIC of 3910 and a LOO of 3911.6, which are almost identical to the original values).

Note also that the same uniform distribution is used in the BCP model. In both models, the choice of a uniform distribution (used also by Wilson & Niv [19]) is not coincidental. It is used to define $h$ as transition uncertainty rather than volatility. That is, when $h = 1$, the agent ignores previous knowledge completely, learning anew on each trial, rather than being certain that the previous cue is no longer relevant.

## Model fitting procedure

Model parameters were estimated in a hierarchical Bayesian framework using Stan [47,48] which implements Hamiltonian Markov Chain Monte Carlo sampling. The free parameters of the model ($h$, $\gamma$, $\epsilon$) are parameterized as probabilities ($h$ and $\epsilon$ range from 0 to 1, whereas $\gamma$ ranges from 0.5 to 1), which in Stan are commonly modeled using an inverse-probit transformation (or an approximation thereof using the Stan Phi_approx function [47]) of normally distributed numbers. Moreover, in Stan it is usually recommended to parameterize hierarchical models using a *non-centered parameterization*, which improves the sampling process, by sampling from independent standardized normal distributions, and transforming the sampled parameters to construct the hierarchy (instead of sampling participant-level parameters directly from the group-level distribution; [47,49]). A graphical representation depicting the dependencies between the different parameters of the hierarchical model used for estimation can be found in Fig 6.

In Fig 6 and below, parameters with an $s$ subscript correspond with participant-level parameters. Parameters ending with 0 correspond with the continuous, normally distributed parameters which were (inverse-probit) transformed to create the uncertainty parameters used in the learning models above.

The following set of equations defined the relationship between the different parameters and auxiliary parameters:

$$h_s = \phi(\sigma_{h0}h0_s + \mu_{h0}) \tag{27}$$

$$\gamma_s = \frac{\phi(\sigma_{\gamma0}\gamma0_s + \mu_{\gamma0})}{2} + 0.5 \tag{28}$$

where the division by 2 and addition of 0.5 limits $\gamma_s$ from 0.5 to 1, instead of 0 to 1, and:

$$\epsilon_s = \phi(\sigma_{\epsilon0}\epsilon0_s + \mu_{\epsilon0}) \tag{29}$$

In such a non-centered parameterization, the standard normal distribution was used as a prior for all auxiliary participant-level parameters ($h0,\gamma0,\epsilon0$). In addition, we used the standard normal distribution as a hyperprior for group-means ($\mu_{h0},\mu_{\gamma0},\mu_{\epsilon0}$). Due to the probit transformation, this is mathematically equivalent to setting a uniform (thus, non-informative) prior on $h$, $\gamma$ and $\epsilon$ at the group level. Finally, for hyperpriors on the group standard deviations ($\sigma_{h0}$, $\sigma_{\gamma0},\sigma_{\epsilon0}$) we used the half-$t$ distribution with a mean of 0, a standard deviation of 0.2, and a $v$ parameter of 50. This produces a uniform prior for individual-level parameters (as similar prior was implemented, probably due to similar reasons, in the hBayesDM package [50]), yet the use of a half-$t$ distribution (instead of a half-normal distribution) allows for some variation between participants even in the case in which group-level parameters are extreme (e.g., close to 1).

For each model, the MCMC was run in three chains, with 1000 samples in total, 400 of which were used during warmup to calibrate the Hamiltonian parameters, and were discarded (each of the models required several days to run, and therefore increasing the MCMC samples to considerably larger numbers was impractical). To ensure unbiased sampling, for models in

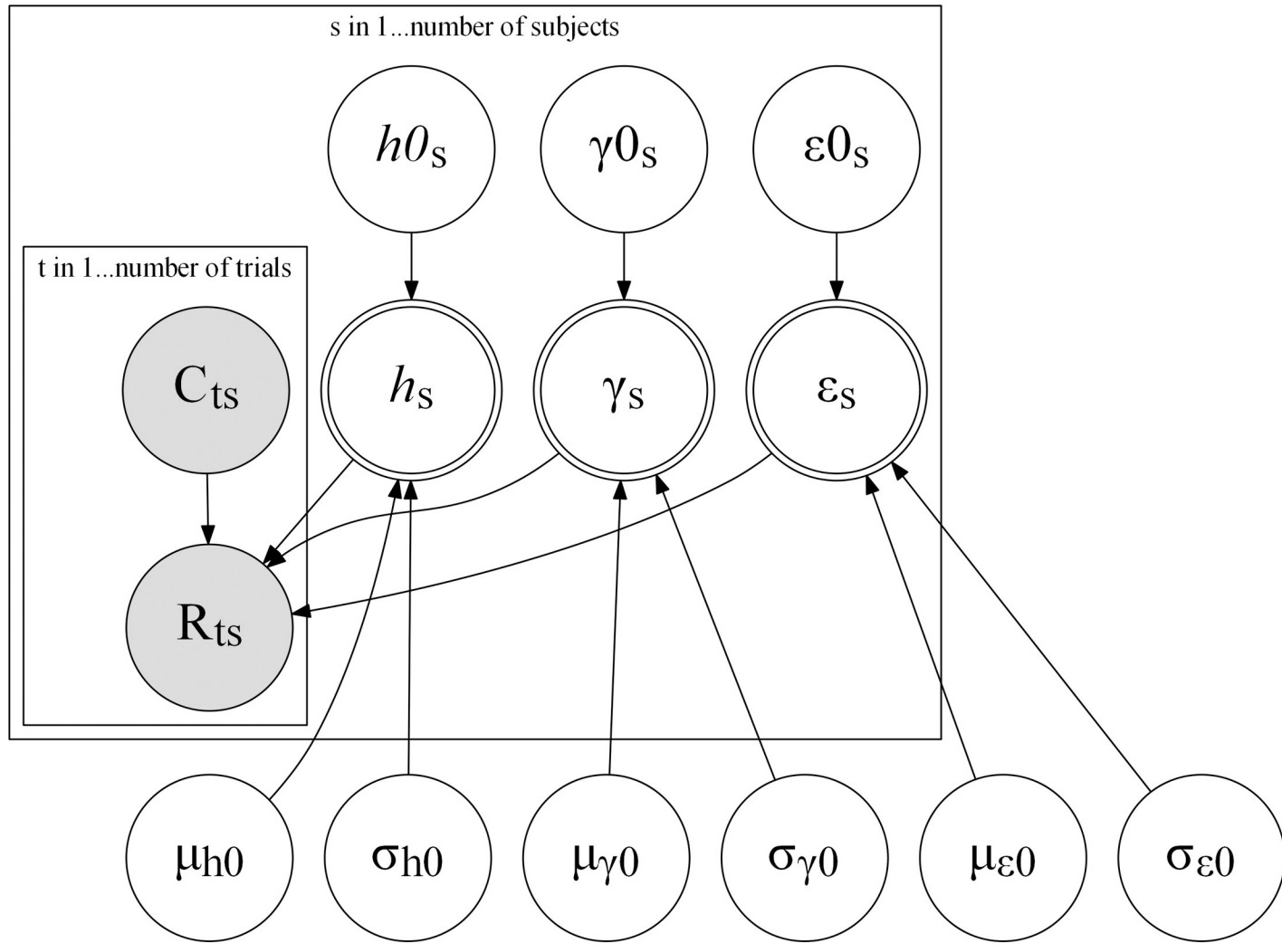

**Fig 6. Graphical representation of the hierarchical Bayesian model used for the estimation of parameters in the BCP models.** A similar model was used for the SA models, with the addition of the $\lambda_0$ parameter. Shaded circles denote observed variables (cues and responses), blank circles denote latent variables, and double-circles represent variables that have deterministic relationships to other variables in the model. Nodes inside the rectangular plate are modeled for each individual participant or trial within participants.

which divergent transitions were detected (typically 1 or 2 transitions), the MCMC acceptance rate parameter (adapt_delta) was gradually increased (up to a maximum of 0.99)–which eliminated all divergent transitions. All models converged as indicated by $\hat{R}$ values the did not exceed 1.1. To improve the accuracy of individual-level parameters for the examination of the main hypotheses, the best-fitting models were run again for a larger number of iterations (1500 samples per chain, with 500 warmup samples).

## Questionnaires

**Obsessive-compulsive inventory-revised [17].** The OCI-R is an 18-item self-report measure of OCD symptoms. It has demonstrated sound psychometric properties in clinical and student populations [51,52]. OCI-R scores in our sample ranged from 0 to 50, with ~40% of participants scoring above 21, which is a common clinical cutoff [17], and ~18% scoring above 30.

**Depression anxiety and stress scale-21 [53].**   The DASS-21 is a reliable and valid measure of anxious arousal (range in our sample = 0–18), stress (range = 0–20) and negative affect (range = 0–19). These scales were used as controls to test the specificity of our findings to OCD symptoms.

## Statistical models for data analysis

Logistic multilevel models (using the lme4 package [16]) were used to investigate the variables affecting accuracy. The models included several trial-level variables (e.g., block, trial) and OCI-R score as a participant-level variable. All models included a random intercept and a random slope when relevant (i.e. analyses including trial-level variables). In models examining an interaction effect (e.g., the interaction of OCI-R and block) all variables were centered before analysis. Analyses were run separately for the probabilistic and deterministic blocks. All models converged.

Analyses involving the prediction of information-theoretic measures of prediction errors (i.e. surprisal and KL divergence) involved linear multilevel models, with random intercept and random slopes for all trial-level variables (all models converged, here approximate $p$-values were calculated by using the lmerTest package [54]). Finally, correlations between fitted parameters (taking the median of each posterior distribution) and participants' scores on the OCI-R and the DASS-21 were tested by using a permutation test, due to the violation of the normality assumption for most variables [55].

## Supporting information

**S1 Text. Procedure and results of the spatial cueing response times task (Posner task).**
(DOCX)

**S2 Text. Discussing the full model comparison results.**
(DOCX)

**S1 Table. Effects for the BCP model with changing (vs. constant) h .**
(DOCX)

**S2 Table. Full model comparison results.**
(DOCX)

## Author Contributions

**Conceptualization:** Isaac Fradkin, Jonathan D. Huppert.

**Formal analysis:** Isaac Fradkin, Casimir Ludwig, Eran Eldar.

**Funding acquisition:** Jonathan D. Huppert.

**Investigation:** Isaac Fradkin.

**Methodology:** Isaac Fradkin, Casimir Ludwig, Eran Eldar, Jonathan D. Huppert.

**Project administration:** Isaac Fradkin.

**Resources:** Jonathan D. Huppert.

**Software:** Isaac Fradkin.

**Supervision:** Jonathan D. Huppert.

**Visualization:** Isaac Fradkin.

**Writing – original draft:** Isaac Fradkin.

**Writing – review & editing:** Isaac Fradkin, Casimir Ludwig, Eran Eldar, Jonathan D. Huppert.

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
