## [Decision Letter · Decision Letter 0]

28 Sep 2019

Dear Dr Fradkin,

Thank you very much for submitting your manuscript 'Doubting what you knew and checking: uncertainty regarding state transitions is associated with obsessive compulsive symptoms' for review by PLOS Computational Biology. Your manuscript has been fully evaluated by the PLOS Computational Biology editorial team and in this case also by independent peer reviewers. The reviewers appreciated the attention to an important problem, but raised some substantial concerns about the manuscript as it currently stands. While your manuscript cannot be accepted in its present form, we are willing to consider a revised version in which the issues raised by the reviewers have been adequately addressed. We cannot, of course, promise publication at that time.

Sincerely,

Samuel J. Gershman

Deputy Editor

PLOS Computational Biology

[LINK]

Reviewer's Responses to Questions

**Comments to the Authors:**

Reviewer #1: Thank you for the opportunity to review this manuscript, which reports an interesting behavioural study of the association between obsessive-compulsive symptoms and state transition uncertainty in the context of a multi-cue reversal learning task. The study is innovative, methodologically sound, and well-written in general (though see several points below). Strengths of the manuscript include clear and comprehensive presentation of a complex set of computational models, the use of cutting-edge methods for Bayesian model estimation using Hamiltonian Monte Carlo.

Although I found the manuscript to be rigorous and appropriately framed in general, I did have several issues that I feel should be addressed. I believe that addressing these concerns will further enhance the presentation of these findings.

#### Major points

1. As a general point, given that the data for this manuscript are to be made freely available in an Open Science Framework repository, it would be useful to also provide the Stan code for each model.

2. As I understand it, the response model for the Bayesian Change Point models assumes that the subject produces a response by marginalising over the different cues (i.e., weighting each cue according to the probability that it is valid). However, this assumption has not been empirically tested. Given that only a single cue is valid on any given trial, it is plausible that participants might respond only according to the cue with the highest probability of being valid (rather than marginalising across all cues). For instance, if my belief is that cue A has a 70% probability of being the valid cue, B a 10% probability, and C a 20% probability, my response might be determined according to the direction of cue A alone (at least in an epsilon-greedy fashion), rather than 0.7 x A + 0.1 x B + 0.2 x C. I would be interested to see this model tested, since it represents a kind of intermediate step between the BCP models (which track the probability of all cues simultaneously) and the SA models (which follow an epsilon-greedy policy for the attended cue).

3. The manuscript claims (e.g., page 18, line 364) that the observed results are specific to OC symptoms, and not symptoms of depression or anxiety, but the evidence for this specificity is weak. As far as I can tell, the specificity claim rests on the finding that the transition uncertainty parameter was significantly correlated with OC symptoms (r = 0.33 and 0.26), but that correlations between the transition uncertainty parameter and depression and anxiety symptoms were not statistically significant (r < .24, p < .1). This evidence is not sufficient to demonstrate that findings are specific to OC, given that, the difference between the statistically significant OC results and the non-significant anxiety and depression results may not itself be statistically significant. To test this claim, it is necessary either to test whether the OC symptoms correlated with transition uncertainty *more strongly* than depression and anxiety symptoms (e.g., using a Fisher r-to-z transformation of the observed correlations), or to show that the OC result was significant even when depression and anxiety were controlled for. If the absence of these findings, there is no strong evidence for the specificity of observed results to OC symptoms, and statements to this effect (e.g., the final sentence of the first Discussion paragraph) should be re-written or removed, and the possibility of confounding by other psychiatric symptoms should be discussed.

4. The manuscript states that increased transition uncertainty may explain checking behaviours in OCD. This is a reasonable speculation, but it is important to note that this study only tested this association indirectly. Although the transition uncertainty parameter did correlate positively with self-reported checking, it also correlated equally strongly with symptoms of Neutralizing, Hoarding, and Washing. I understand that the latter analyses were exploratory, but nevertheless it is overstating findings to say that transition uncertainty "can explain excessive checking" (page 21, line 426) given that no specificity to checking behaviours was observed. I feel that this ought to be noted in a revised manuscript. I would also suggest for the same reasons that the "and checking" portion of the manuscript title is a significant overstatement of the results of the manuscript.

5. As it currently stands, the manuscript tests two families of models: a Bayesian optimal observer family and a family based on the Yu & Dayan approximation. Both are interesting and reasonble, but testing only these two families of models implicitly assumes that participants are not employing some simpler strategy. In order for the results of this manuscript to be interpretable, it is imperative to rule out the possibility that participants are employing a simpler strategy. I would suggest testing a simpler model to rule it out as a competing account of behavioural data: a win-stay lose-switch model that assumes that participants initially pick a cue at random, and then (probabilistically, according to a parameter that varies across participants) stick with that cue if it correctly predicts the outcome, or switch to another cue (again, probabilistically according to a fitted parameter) if their chosen cue does not correctly predict the outcome.

6. The manuscript interprets the primary correlation between the transition uncertainty parameter and OC symptoms in terms of increased uncertainty concerning state transitions. According to the model framework presented, the source of this increased uncertainty is a generative model of the environment as more volatile. However, transition uncertainty may have other psychological sources, and it would be useful for the manuscript to discuss these in more depth. For instance, if participants do not maintain Bayesian belief updates but recompute probabilities on the fly, poorer memory recall of previous outcomes would similarly result in an increased reliance on the most recent outcomes. Similarly, lack of confidence in one's own memory for previous outcomes (see, e.g., Boschen & Vuksanovic, Behaviour Research and Therapy, 2007; Tolin et al., Behaviour Research and Therapy, 2001) would lead one to have greater uncertainty about state transitions, but not because one believes that the environment is more volatile. Similarly, in the domain of checking behaviours the manuscript would benefit from engaging with and discussing previous cognitive models of checking that have been proposed in the literature. For instance, it has been proposed (see, e.g., Lind & Boschen; Journal of Anxiety Disorders, 2009; Bennett et al., PLoS Computational Biology, 2016) that checking in OCD results not from increased uncertainty but from increased aversion to uncertainty when it is present, and therefore greater relief from anxiety by checking behaviours.

7. It would be useful to provide more information on the details of the Stan sampling procedure. How many samples were taken in total, and how many were discarded during the warm-up phase? How many chains were used to sample from the posterior, and did these chains converge (as indiciated by the R_hat statistic)? Were there any divergent transitions in any model?

#### Minor points

Abstract: The acronym 'OCD' is not introduced here or elsewhere.

Page 5, lines 96-97: Given that the Yu & Dayan model is rather complex (with links to attention, learning, and neuromodulation, as the manuscript notes), it might clarify things to expand on precisely which of its features have not been empirically examined.

Page 5, lines 105-106: Citations should be given for the attribution of this "common preconception" regarding OCD.

Page 6, line 117: I believe that "which allows to independently quantify" is ungrammatical. "which allows us to independently quantify" would be correct, as would "which allows independent quantification of".

Page 6, line 129: The exact information concerning the timing of the shift is not provided until the Methods section, but it would be useful for the reader in interpreting the results presented in this section to know that there was only one shift per block, in roughly the middle of the block.

Page 7, Figure 1: '40/48' somewhat implies that the transition occurred after either 40 or 48 trials. 40-48 would be more accurate.

Page 7, section beginning line 142: I feel that it would be useful for the reader to know more basic behavioural descriptive statistics before the manuscript jumps directly to the inferential analyses. For instance, on what proportion of trials did participants respond correctly? How did this change over the course of the pre-shift and post-shift blocks? Were participants at asymptote at the time the shift occurred, and if so, what what the asymptotic level of performance?

Page 7, line 147: The acronym 'OCI-R' is used before it is defined.

Page 12, line 256: Typo - Criterion, not Criteria.

Page 15, line 15: Several questions about these confidence intervals: What percentage confidence interval is presented in this figure? Are the confidence intervals across subjects or trials? Is this a Bayesian highest density interval (in which case, it would be more appropriate to call it a credible interval) or a frequentist confidence interval? If the latter, what is the rationale for mixing Bayesian and frequentist statistics? I applaud the manuscript's efforts to present an estimate of uncertainty around model fit statistics, but I think a little more transparency would be useful.

Page 20, line 403: Typo - "this task differ".

Page 22, line 463: "After a random number of 40-48 trials". I assume that integers were chosen from this range according to a uniform distribution? It would be helpful to have that information explicitly reported in the manuscript.

Page 22, line 463: At the switch-point, was the new predictive cue guaranteed to be different from the previously predictive cue, or was the new predictive cue equally likely to be any of the three cues? I assume the former, but it would be helpful to have this information explicitly reported in the manuscript (especially since the models appear to assume the latter).

Page 34, lines 700-701. "All models included a random intercept and a random slope when relevant". Did all models converge? If a model did not converge, what protocol was used for simplifying the model? Also: given that the lme4 package does not produce p-values, please provide information on which approximation was used to estimate these.

Supplementary material, Table S2: According to what criterion were models inferred to be significantly different from one another (alphabetical superscripts)?

Reviewer #2: This is a timely and important paper. It employs a Computational Psychiatry approach to elucidate mechanisms behind clinically observed symptoms of obsessive-compulsive disorder (OCD). Emergent literature converges in understanding that there are significant impairments in decision making among individuals with OCD even in contexts unrelated to their main symptomology. Including computational modeling approach in broader investigations is a more effective strategy to explore such impairments than clinical observations alone, since it allows testing empirically hypotheses that are largely based on clinical intuition. The results are interesting and are with many implications for further research.

The major challenge of papers like this is to make the tools and results more accessible to clinical audience. The current draft falls a bit short of that. It is a VERY technical paper. It is not an easy read for people without computational background. I suggest this is a main goal for the next revision.

Maybe, state two alternative hypotheses (H0: Reduced transition uncertainty, Ha: Inflexibility in strategies) very clearly from the start and explain how clinical observations may support each of them. Then list all relevant quantitative variables with clear intuitive explanation of what they mean, and how using these variables may test H0 against Ha. This will also reinforce the comparative strength of using computational approaches – an ability to empirically test alternative hypotheses. But limitations (and how they may relate to the assumptions) should be discussed as well. I think that the models should not be described in such details in the main text and could be left more for Methods Section. But what is needed is an added value of this approach, what is predicted and can be tested in very simple intuitive terms, why it is important to use multiple models to test these predictions, and why it is important to have both probabilistic and deterministic conditions – again, a value added in a very plain language. Then present all results in the simple language as well – consistent with this hypothesis and rejects this hypothesis + exploratory analyses (maybe some of the exploratory results could be moved to supplements – there are so many of them, may be try to prioritize?). This or another strategy to present the results in more clinically relevant manner is likely to improve the impact of this research.

Specific comments:

Major:

There is a LARGE number of correlational results reports in the paper. This raises a concern of the multiple comparisons problem. Significance levels are unlikely to survive the corrections. Furthermore, the data appears to be subject to increasing variance (vs. symptom severity) and not necessarily normally distributed. Maybe it is more appropriate to use nonparametric correlational analyses here. Yet, the more powerful approach would be to modify the model to allow individual variations in key parameters (e.g. hi ~ h * OC-R). This would resolve both concerns. Would it be possible to fit such model given the data?

Minor:

p. 10. “where (+1|,1:) reflects the prior probability that the previously relevant cue is no longer relevant on trial t+1 (i.e., transition uncertainty).”

- you probably meant “the probability that this experience is still/no longer relevant (i.e., in case of a changepoint)”, as you correctly state on p. 24.

p. Fig. 3 legend “additional evidence for negligent, chance-level performance.”

- performance based exclusion criteria should be stated before the data/results. The way it is presented it sound post hoc.

p. 21. “The next step is to develop a more ecological design, examining whether transition uncertainty can explain actual OC symptoms.”

- this sounds contradictory to the presented motivations (general cognitive impairments) and correlational analyses with OC severity. Probably meant to test the model in clinically relevant contexts/symptom provocation paradigms.

p. 21. “Next, we examined the specificity of transition uncertainty to OC symptoms. In support of specificity, transition uncertainty was not significantly correlated with anxious arousal, depressive symptoms or stress…”

- Need to include the rationale for this before data are presented. As is it shows up from nowhere.

**Have all data underlying the figures and results presented in the manuscript been provided?**

Reviewer #1: No: Although the raw data have been provided in an OSF repository, numerical data underlying graphs and summary statistics have not been provided in spreadsheet form as supporting information.

Reviewer #2: Yes

PLOS authors have the option to publish the peer review history of their article (what does this mean?). If published, this will include your full peer review and any attached files.

Reviewer #1: No

Reviewer #2: No

---

## [Decision Letter · Decision Letter 1]

6 Jan 2020

Dear Dr Fradkin,

We are pleased to inform you that your manuscript 'Doubting what you already know: uncertainty regarding state transitions is associated with obsessive compulsive symptoms' has been provisionally accepted for publication in PLOS Computational Biology.

In the meantime, please log into Editorial Manager at https://www.editorialmanager.com/pcompbiol/, click the "Update My Information" link at the top of the page, and update your user information to ensure an efficient production and billing process.

One of the goals of PLOS is to make science accessible to educators and the public. PLOS staff issue occasional press releases and make early versions of PLOS Computational Biology articles available to science writers and journalists. PLOS staff also collaborate with Communication and Public Information Offices and would be happy to work with the relevant people at your institution or funding agency. If your institution or funding agency is interested in promoting your findings, please ask them to coordinate their releases with PLOS (contact ploscompbiol@plos.org).

Thank you again for supporting Open Access publishing. We look forward to publishing your paper in PLOS Computational Biology.

Sincerely,

Samuel J. Gershman

Deputy Editor

PLOS Computational Biology

Reviewer's Responses to Questions

**Comments to the Authors:**

Reviewer #1: I thank the authors for their thorough revision, which has addressed all of my concerns.

**Have all data underlying the figures and results presented in the manuscript been provided?**

Reviewer #1: Yes

PLOS authors have the option to publish the peer review history of their article (what does this mean?). If published, this will include your full peer review and any attached files.

Reviewer #1: No

---

## [Editor Report · Acceptance letter]

28 Jan 2020

PCOMPBIOL-D-19-01377R1 

Doubting what you already know: uncertainty regarding state transitions is associated with obsessive compulsive symptoms

Dear Dr Fradkin,

I am pleased to inform you that your manuscript has been formally accepted for publication in PLOS Computational Biology. Your manuscript is now with our production department and you will be notified of the publication date in due course.

With kind regards,

Laura Mallard
